# The Interplay Between Immunity, Inflammation and Endothelial Dysfunction

**DOI:** 10.3390/ijms26041708

**Published:** 2025-02-17

**Authors:** Ying Jie Chee, Rinkoo Dalan, Christine Cheung

**Affiliations:** 1Department of Endocrinology, Tan Tock Seng Hospital, Singapore 308433, Singapore; rinkoo_dalan@ttsh.com.sg; 2Lee Kong Chian School of Medicine, Nanyang Technological University, Singapore 308232, Singapore; ccheung@ntu.edu.sg; 3Institute of Molecular and Cell Biology, Agency for Science, Technology and Research, Singapore 138632, Singapore

**Keywords:** endothelium, cardiovascular, immunity, inflammation, therapeutic agents

## Abstract

The endothelium is pivotal in multiple physiological processes, such as maintaining vascular homeostasis, metabolism, platelet function, and oxidative stress. Emerging evidence in the past decade highlighted the immunomodulatory function of endothelium, serving as a link between innate, adaptive immunity and inflammation. This review examines the regulation of the immune–inflammatory axis by the endothelium, discusses physiological immune functions, and explores pathophysiological processes leading to endothelial dysfunction in various metabolic disturbances, including hyperglycemia, obesity, hypertension, and dyslipidaemia. The final section focuses on the novel, repurposed, and emerging therapeutic targets that address the immune–inflammatory axis in endothelial dysfunction.

## 1. Introduction

The vascular endothelium is a single-cell layer that lines the surface of blood vessels, serving as the primary point of contact with the circulation and facilitating the exchange of nutrients and metabolites [1]. A healthy, quiescent endothelium is crucial for maintaining vascular homeostasis, as it regulates the synthesis of nitric oxide (NO) and other vasodilators, such as prostacyclin and other endothelium-derived hyperpolarizing factors [2].

In the presence of atherogenic stimuli, the quiescent endothelial cells (EC) become activated. The hallmarks of endothelial dysfunction, as described by Hunt and Jurd, encompass the following: impaired vascular integrity exposing the subendothelium, increased expression of adhesion molecules facilitating transendothelial leucocyte movement, prothrombotic tendency, inflammation, and the expression of major histocompatibility complex (MHC) II on ECs, enabling them to function as antigen presenting cells (APCs) [3].

Endothelial dysfunction is one of the earliest manifestations of atherosclerosis [4]. The progression of atherosclerosis occurs over a relatively long, asymptomatic interval and may begin as early as childhood [5]. Over 70% of young adults develop early signs of atherosclerosis, such as fatty streaks in the arterial walls [6]. Early in the atherosclerotic process, the endothelium becomes activated upon exposure to atherogenic stimuli and serves as a focal point for initiating the atherosclerotic process, particularly at artery bifurcations, where turbulent blood flow increases endothelial shear stress [7]. Endothelial activation upregulates adhesion molecules, accelerates the release of proinflammatory cytokines, and promotes leukocyte migration into the subendothelial space [8]. Within the arterial intimal layer, recruited monocytes transform into macrophages that phagocytose apolipoprotein (apo) B particles to form foam cells [8]. The accumulation of foam cells leads to the development of fatty streaks and plaque formation in the arterial intima. Vascular smooth muscle cells then migrate to the atherosclerotic plaque, forming a fibrous cap that degrades over time, ultimately leading to plaque rupture [9].

Traditionally, the role of the immune system in atherosclerosis has been attributed to immune cells derived from myeloid and lymphoid stem cells [10]. However, recent evidence suggests that other cell types, such as ECs, are able to modulate immune processes [11]. This review explores the interplay between the immune and inflammatory axis on endothelial dysfunction and atherosclerosis. We examine the immune functions of ECs in physiological, pathophysiological states, and discuss the potential therapeutic strategies that target the immune–inflammatory axis to ameliorate endothelial dysfunction and atherosclerosis.

## 2. Interactions Between ECs and the Innate Immune System

The innate immune system generates immediate immune responses, classically to pathogens by recognizing highly conserved molecular structures known as pathogen-associated molecular patterns through pattern recognition receptors (PRRs) [12,13]. Innate immune responses can also be triggered in the absence of pathogens. In this context, sterile inflammation is induced by endogenous metabolites known as damage-associated molecular patterns (DAMPs) that bind to specific receptors to initiate the inflammatory process [14]. Recently, a subset of immunostimulatory molecular patterns associated with harmful lifestyle factors was classified as lifestyle-associated molecular patterns (LAMPs) [14]. Endogenous LAMPs include cholesterol crystals and urate crystals, which bind to nucleotide-binding domain, leucine-rich-containing family, pyrin domain-containing-3 (NLRP3) [15,16], CD36 receptors [17], while oxidized low-density lipoprotein (oxLDL) binds to toll-like receptor (TLR) 4 [18,19], CD36 [20] and lectin-like oxLDL (LOX-1) receptors expressed on macrophages and ECs [21] to activate downstream inflammatory responses, including nuclear factor κB (NF-κB), NLRP3 inflammasome and the secretion of interleukin (IL)-1β [22].

ECs possess a characteristic feature of innate immune cells by releasing a diverse array of cytokines to various stimuli. These include proinflammatory cytokines such as IL-1β, which is generated through the activation of the NLRP3 inflammasome [22], as well as IL-3, IL-5, IL-6, IL-8, monocyte chemoattractant protein 1 (MCP-1) and tumour necrosis factor alpha (TNF-α) [23]. Furthermore, adhesion molecules, including P-selectin, E-selectin, vascular cell adhesion molecule-1 (VCAM-1) and intracellular adhesion molecule-1 (ICAM-1), are upregulated in activated ECs, which act synergistically with proinflammatory cytokines to promote leukocyte recruitment [24]. Neutrophils are typically the first immune cells to arrive at sites of inflammation [25]. In a series of coordinated steps, neutrophils tether and roll on the endothelial surface [26] before transendothelial migration [27] into the subendothelial space.

Monocytes are recruited to the arterial intima in a similar mechanism to neutrophils. In a tightly regulated cascade, endothelial P-selectin and E-selectin facilitate the tethering and rolling of monocytes, with integrins enhancing the binding affinity and firm adhesion of monocytes to VCAM-1 [28]. Activated ECs express chemokine receptors, for example, chemokine (C-C motif) receptor-like 2 (CCLR2), C-C chemokine receptor type 5 (CCR5) and CXC chemokine receptor 2 (CXCR2), which bind to the respective chemokine ligands to promote monocyte trafficking [29,30]. Activation of integrins by chemokines ensures monocytes firmly adhere to the endothelium. Subsequent polarization changes monocyte configuration, facilitating their transmigration into the subendothelial space [30]. Human monocytes are identified via CD14 and CD16 on the cell surface, categorizing them into classical (CD14++CD16−), intermediate (CD14++CD16+), and non-classical (CD14+CD16++) monocytes [30]. In the intima, monocytes differentiate into proinflammatory M1 macrophages and phagocytose oxidized lipids to form foam cells [31]. The proliferation of macrophages is mediated by a granulocyte macrophage-stimulating factor, which promotes plaque growth [32].

The complement system plays a pivotal role in the innate immune system. There are three major pathways in the complement cascade—classical, alternative, and mannose-binding lectin (MBL) pathways. The classical pathway is initiated by the binding of immune complexes to C1q [33]. The alternative pathway begins with C3 cleavage to C3b, while the MBL pathway is activated when the MBL protein binds to mannose residues on pathogens [33]. The three pathways converge to a shared pathway, ultimately forming a membrane attack complex (MAC) [33]. MAC induces MCP-1 [34], adhesion molecules [35], promotes leukocyte migration [36], and enhances tissue factor synthesis to generate prothrombotic states [37]. There are conflicting data regarding the effects of different complement factors on atherosclerosis. While cross-sectional studies showed positive associations between circulating complements such as plasma C5 [38] and C5b-9 with endothelial dysfunction and subclinical atherosclerosis [39], C1q and C3 deficiencies aggravate atherosclerosis, presumably due to the role of C1q and C3 in promoting apoptotic cell clearance [40]. To protect against the atherogenic effects of MACs, complement regulatory proteins, such as CD55 and CD59, inhibit MAC formation [33]. Complement factor H, a key regulator of the alternative complement pathway, is upregulated in monocytes and macrophages and its deficiency significantly reduces atherosclerotic plaque size [41]. As a mediator of the C5b-9 pathway, the response gene to complement 32 (RGC-32) protein, which is predominantly expressed in ECs, plays a crucial role in atherogenesis. RGC-32 deficiency attenuates monocyte-EC interactions and adhesion molecule expression by disrupting the NF-kB pathway [42].

Readers may refer to [11] for an in-depth discussion of the innate immune functions of ECs.

## 3. ECs as a Bridge Between Innate and Adaptive Immunity

Endothelial–leukocyte interactions are key in bridging the link between immunity, inflammation and endothelial dysfunction. Lining the apical surface of the endothelium, the endothelial glycocalyx (EG) is a thin protective barrier that can be readily damaged by various inflammatory stimuli [43]. The EG is composed of proteoglycans, mainly syndecans, glypican-1, which are, in turn, bound to glycosaminoglycans, predominantly heparan sulfate, chondroitin sulfate, and hyaluronic acid [43]. Excessive exposure to inflammatory stimuli such as glucose [44], salt [45], turbulent blood flow [46], and proinflammatory cytokines such as TNF-α [43]) can induce EG degradation, a process known as ‘endothelial shedding’. This disruption not only compromises the glycocalyx barrier, increases endothelial permeability to harmful mediators [43], but also generates glycocalyx fragments that are capable of binding to TLRs to trigger further cytokine release [47]. Thinning of the EG increases its permeability to promote the infiltration of neutrophils and macrophages, which can further activate glycocalyx-degrading enzymes such as hyaluronidase and metalloproteinase [48]. Glycocalyx fragments, including syndecan [49], endocan [50], and hyaluronan [51] are measurable in the circulation. These markers provide a readout of endothelial barrier damage and correlate with vascular complications.

Another characteristic of ECs in mediating the endothelial–leucocyte interactions lies in the ability to present antigens to CD4+ and CD8+ T cells through MHC I and II [11]. In contrast to professional APCs that can activate naive T cells, ECs lack the co-stimulatory ligands CD80 and CD86 that are necessary to induce naive T cells [52]. Therefore, ECs can only activate T effector memory cells with prior exposure to the presented antigens [52].

Effective antigen presentation by ECs to T cells requires specific interactions between EC surface ligands and corresponding costimulatory receptors on T cells [52]. The CD40-CD40 ligand (CD40-CD40L) interaction, which involves the binding of CD40 on endothelial cells to its corresponding receptor CD40L on T cells, generates costimulatory signals to activate adaptive immunity [53]. This upregulates endothelial VCAM-1, ICAM-1 and E-selectin to facilitate lymphocyte migration [52]. Furthermore, the CD-40-CD40L interaction reduces endothelial nitric oxide synthase (eNOS) expression and endothelium-dependent vasodilation in human coronary artery ECs [54]. In atherosclerotic plaques, CD40-CD40L interactions promote the formation of a necrotic core and increase plaque vulnerability [55]. Another costimulatory interaction involves the binding of OX40L on endothelial cells to the corresponding receptor OX40, which enhances T cell proliferation and synthesis of cytokines IL-2, IL-3, and interferon-gamma (IFN-γ) [56]. Conversely, there are certain types of endothelial-T lymphocyte interactions that are atheroprotective. For example, CD2-CD58 co-stimulation promotes regulatory T (Treg) cell differentiation and augments IL-2 and IL-10 production [57]. Another example is the binding of programmed cell death ligand 1 (PD-L1) expressed on endothelial cells with programmed death protein 1 (PD-1) on T cells, which suppresses T-cell mediated cytotoxicity and induces T cell apoptosis [58].

The trafficking of T lymphocytes into the subendothelial space is similar to innate immune cells. The recruitment cascade begins with T cell adhesion to the endothelium mediated by chemokines that activate integrins [59]. Subsequently, distinct patterns of T cell rolling on the activated endothelium are facilitated by interactions between various combinations of selectin ligands with the ECs [59]. Specific chemokines, such as chemokine ligand 20 (CCL20) promote T helper (Th) 17 cell adhesion [60], while CXC motif chemokine ligand (CXCL) 9 and CXCL10 enhance T cell binding to endothelial ICAM-1 [61]. Diapedesis of T cells is mediated by interactions between P-selectin and VCAM-1 on ECs with lymphocyte function-associated antigen 1 (LFA-1) and α4β1 integrin (VLA-4) on T cells, respectively [62]. Finally, the binding of ICAM-1 [63] and VCAM-1 on ECs to T cells facilitates the transient dissociation of EC cadherin, creating a gap to enable the transendothelial migration of T cells [64].

Emerging evidence suggests that activated ECs can induce T-cell CD69 expression independent of T-cell receptor (TCR) stimulation. This alternative mechanism is facilitated by IL-5, VCAM-1, and ICAM-1. The activation of ECs leads to the sustained upregulation of CD69 expression on T cells, providing a novel means of priming T cells for residency and driving transendothelial migration [65].

To maintain immune homeostasis, T cells can differentiate into a small subset of regulatory T cells (Tregs), a process partly regulated by ECs. Tregs are characterized by transcription factor forkhead box protein 3 (FOXP3), IL2 receptor subunit alpha, and cytotoxic T lymphocyte-associated protein 4 [66]. Tregs counteract chronic inflammation by secreting immunosuppressive cytokines, including IL-10, IL-35 and transforming growth factor-β (TGF-β), which stabilizes atherosclerotic plaque [67]. The endothelium compensates for the increase in proinflammatory T effector lymphocytes with an augmentation of Treg function, as demonstrated by the induction of Treg proliferation when IFN-γ-stimulated human umbilical vein ECs (HUVECs) and dermal ECs were co-cultured [67]. A stable expression of FOXP3 is crucial for maintaining Treg function, as preclinical models demonstrated that the loss of FOXP3 attenuated the immunosuppressive effect of Tregs [68]. This loss of Treg function could be attributable to the reduction in FOXP3 expression as excess cholesterol accumulates [66]. Furthermore, a proportion of Tregs are converted into proatherogenic T cells, shifting the initial protective function of Tregs to pathogenic as atherosclerosis progresses [69].

The role of antibodies and endothelial dysfunction:

There is a clear relationship between autoimmune diseases and accelerated atherogenesis [70]. While autoimmune disease-related inflammation is a major contributor to endothelial dysfunction, the formation of autoantibodies, unique to the pathogenesis of the rheumatic disease, could be another mechanism that accounts for the similarities in rheumatic disease manifestations and vascular complications [71]. The anti-endothelial cell antibodies (AECAs) belong to a heterogeneous group of antibodies that interact with EC-bound antigens [72]. AECAs are highly prevalent in autoimmune diseases and are detectable in up to 70% of individuals with rheumatoid arthritis and 90% in systemic lupus erythematosus [72]. In addition, AECAs have been detected in non-immune atherosclerotic diseases such as peripheral vascular disease [73] and post-acute myocardial infarction [74]. AECAs upregulate adhesion molecules, proinflammatory cytokines, and the migration of leukocytes, and they promote thrombogenesis [75,76]. Exposure of endothelial cells to atherogenic stressors triggers the formation of autoantibodies against heat shock protein 65 expressed on the EC surface [77] and antibodies against EC-derived glucose-regulated protein 78 (GPR78) [78], upregulating the expression of adhesion molecules via NF-κB [78]. In recent years, there has been an increasing focus on the role of natural immunoglobulin M (IgM) antibodies in the field of atherosclerosis. These antibodies are atheroprotective by binding to DAMPs or PAMPs to prevent uptake by scavenger macrophages and reducing foam cell formation [79].

The immune–inflammatory interactions in the endothelium are summarized in Figure 1.

In 2023, the Global Cardiovacular Risk Consortium harmonized more than 1.5 million individual level data. The top five modifiable risk factors—body mass index, systolic blood pressure, non-HDL lipoprotein cholesterol, current smoking, and diabetes mellitus account for 53 to 57% of 10-year incident cardiovascular disease [80]. Emerging non-traditional risk factors, such as infectious diseases and environmental factors are increasingly recognized. In the following section, we elaborate on the interplay between immunity, inflammation, and endothelial dysfunction in the context of various metabolic disorders, including hyperglycemia, obesity, hypertension, and dyslipidaemia. We will discuss how emerging risk factors, including a history of certain infectious diseases and environmental exposure, are emerging key players in aggravating endothelial dysfunction.

## 4. Diabetes and Endothelial Dysfunction

Type 2 diabetes mellitus (T2DM) is characterized by chronic hyperglycemia and insulin resistance. Widely recognised as a vascular disease, the long-standing complications of suboptimally controlled T2DM include microvascular and macrovascular complications. Cardiovascular disease is a leading cause of mortality in T2DM [81]. Endothelial dysfunction is one of the earliest manifestations of diabetes-related vascular complications [82].

Hyperglycemia is a proinflammatory state that is characterized by an accelerated release of cytokines IL-6 [83], IL-8 [84], IL-1β [85], and chemokines such as MCP-1 [86]. In addition, hyperglycemia induces oxidative stress by generating reactive oxygen species (ROS) [87]. ROS leads to the uncoupling of eNOS, which stimulates the release of superoxide instead of NO [88]. The superoxide utilizes NO rapidly to form nitrogen peroxynitrite, a potent oxidant, further depleting NO availability and attenuating NO synthesis [89,90]. Another source of endothelial ROS, the advanced glycation end products (AGEs), are increased in hyperglycemia [91]. Excess ROS stimulate proinflammatory gene expression [92] and degrade the EG [43]. The deleterious effect on the endothelial glycocalyx was confirmed by the structural thinning of glycocalyx, measured using high-resolution microscopy [93] and biochemically through measuring circulating EG fragments [51]. In addition, Nrf2, a critical regulator of antioxidative and anti-inflammatory pathways by promoting Th2 cytokines and suppressing Th1 cytokine release, is reduced in individuals with recent-onset T2DM [94]).

Recently, dysfunctional mitochondria were reported to induce mitochondrial ROS and promote endothelial dysfunction [95]. New mitochondria are regenerated through cycles of mitochondrial fusion and fission to form mitochondria networks [96]. In hyperglycemia, increased expression of fission-1-protein alters mitochondria dynamics, disrupts mitochondrial networks, increases the synthesis of mitochondrial ROS, and markedly reduces eNOS activity [95]. Additionally, ROS activate the NF-κB pathway [87], damage the endothelial barrier [89], increase EC permeability, and enhance lipoprotein deposition in the subendothelial space [90].

Endothelial insulin resistance contributes to endothelial dysfunction by attenuating NO synthesis [97]. Insulin binds to its receptor and activates eNOS via the phosphatidylinositol 3-kinase (PI3K)/protein kinase B (Akt) pathway [97]. Impaired endothelial insulin signalling in the endothelium disrupts this pathway and inhibits NO synthesis. Targeting endothelial insulin resistance could potentially ameliorate endothelial dysfunction [97]. Another potential strategy that enhances eNOS activity was reported recently. A novel regulator of eNOS activity, the Takeda G Protein-Coupled Receptor 5 (TGR5), is highly expressed on ECs [98]. TGR5 directly induces eNOS [99] or activates glucagon-like peptide-1 (GLP-1) receptor on ECs to stimulate eNOS activity [100]. Targeting TGR5 could increase NO availability and reverse endothelial dysfunction.

The immune–inflammatory axis is disrupted by hyperglycemia through the following mechanisms. Firstly, hyperglycemia modifies the distribution of circulating immune cells. Individuals with T2DM and coronary artery disease have more proinflammatory Th1 and Th17 cells in the peripheral blood [101], while the anti-inflammatory pool of Th2 and Treg cells is reduced [102,103]. In addition, hyperinsulinemia limits Treg differentiation [104] and reduces the synthesis of anti-inflammatory cytokines IL-10 and TGF-β [104]. Apart from T lymphocytes, hyperglycemia activates neutrophils to release neutrophil extracellular traps (NETs), which are composed of cytoplasmic, granular protein and chromatins [105]. While NETs are critical in counteracting inflammation at physiological concentrations, excess NETs serve as DAMPs to activate the innate immune system [106], upregulate NLRP3 inflammasome [107], and promote thrombosis by directly activating the coagulation system and degrading the anticoagulant tissue factor inhibitor [108].

Secondly, the interactions between PRRs on ECs, B cells, monocytes, and dendritic cells modulate downstream innate immune pathways [109]. Activation of the AGE-RAGE axis and PRRs triggers NF-kB and NLRP3 inflammation formation and increases the secretion of TNF-α, IL6, IL-18, and IL-1β [109,110,111]. Furthermore, AGE-RAGE and TLR4 signalling induces M1 macrophage polarization via the signal transducer and activator of transcription 1 (STAT1) [112] and mitogen-activated protein kinase (MAPK) pathways [113]. In vivo, monocyte counts are elevated in T2DM and promote NLRP3 inflammasome formation, leading to increased synthesis of proinflammatory cytokines IL-1β and IL-18 [110].

Thirdly, a new mechanism highlighting the interactions between insulin, stromal cell-derived factor-1 and its ligand, CXC chemokine receptor 4 (CXCR4), was elucidated in diabetic (db/db) mice. In insulin-resistant ECs, upregulation of CXCR4 increased endothelial leukocyte adhesion and reduced NO synthesis [114].

Another recently described mechanism affecting the immune–inflammatory axis involves the stimulator of interferon genes (STING), a DNA sensor-related protein STING pathway. As a mediator of innate immune signalling, STING, is strongly expressed on vascular cells [115]. STING is activated by cytosolic mitochondrial DNA released in the presence of hyperglycemia and AGEs [116,117]. The cascade of downstream responses of the STING pathway includes activating transcription factors interferon regulatory factor 3 (IRF3), NK-κβ [115,118,119], and reducing eNOS phosphorylation through the Tank-binding kinase-1 (TBK1)-IRF3 pathway [120]. On the contrary, the genetic deletion of STING and STING inhibition ameliorated aortic endothelial inflammation, highlighting the potential role of STING as a therapeutic target in diabetes-related vascular disease [116].

## 5. Obesity and Endothelial Dysfunction

Obesity-induced chronic low-grade inflammation is characterized by increased proinflammatory chemokines, cytokines, adhesion molecules, and endothelin-1 [121,122,123]. The disrupted balance between endothelial vasodilators and vasoconstrictors creates a self-perpetuating cycle that drives endothelial inflammation [124]. Furthermore, the perivascular adipose tissues (PVAT) in obese subjects exhibit impaired vasodilatory response to TNF-α and IL-6 [125], which was reversed by anti-TNF-α monoclonal antibody, infliximab [126], highlighting the role of PVAT in regulating inflammation in obesity.

The immune profiles in obese individuals are altered via a few mechanisms. First, excess saturated fatty acids are recognized by PRRs [127], triggering innate immune pathways, leading to early neutrophilic infiltration [128], loss of protective Treg cells, and enhancing proinflammatory CD8+ T cells in the visceral adipose tissue depot [129]. Emerging evidence highlights the potential involvement of eosinophils in regulating perivascular adipose function, as their depletion in obesity impairs NO release [130].

Adipokines secreted by PVAT are crucial in regulating endothelial function [131]. Leptin derived from healthy PVAT stimulates NO production in the perivascular adipocytes [132]. However, in obesity, dysfunctional PVAT induces leptin resistance and promotes the release of proinflammatory cytokines [133]. Other adipokines that are implicated in obesity, such as visfatin, induce endothelial dysfunction via TLR4-related mechanisms and stimulate NLRP3 inflammasome and IL-1β synthesis [134]. Concurrently, the NO-stimulating protective adipokine, adiponectin, is reduced in dysfunctional PVAT [135].

## 6. Hypertension and Endothelial Dysfunction

The interplay between the immune system and hypertension was first demonstrated in deoxycorticosterone acetate (DOCA) and salt-induced animal models, in which athymic, normotensive mice lacking functional T cells developed hypertension upon the transfer of thymic tissue [136]. Altered innate immune function led to a 3-fold increase in vascular macrophages and the expression of CCR2 and its ligands in hypertensive mice, which was, in turn, reduced by a CCR2 antagonist [137]. Circulating DAMPs, such as salt [138] and sodium urate microcrystals [139], activate macrophages. In vitro culturing of human monocytes with human aortic ECs exposed to varying degrees of cellular stretch displayed enhanced monocyte conversion to intermediate (CD14++/CD16+) monocytes, which was associated with the marked expression of IL-6, IL-1β, IL-23, and TNF-α [140,141]. A key regulator of innate immunity, the high-mobility group box 2 (HMGB2), promotes endothelial injury by interacting with RAGE to increase ROS production [142], leading to arterial stiffness and carotid intima-media thickening [143].

In terms of adaptive immunity, hypertensive mice had increased cytotoxic CD8+ T cell activation in the kidneys [144]. T-cell derived cytokines, particularly IL-17 and IFN-γ, induce angiotensin II production, increase renal tubular sodium reabsorption, impair NO bioavailability, and promote vascular remodelling [145,146,147,148,149]. B-cell involvement was demonstrated by detecting angiotensin II type 1 receptor (AT1R) antibodies in the vessel walls, which interact with angiotensin II to stimulate cytokine production and TGF-β signalling [150]. Individuals with pregnancy-related and essential hypertension express higher AT1R antibodies and respond better to angiotensin receptor blockers, supporting the role of these antibodies as a target in hypertension [151]. In addition, the expression of Fc-γ receptors on ECs allows the binding of immunoglobulin G and inhibiting eNOS [152].

## 7. Dyslipidaemia and Endothelial Dysfunction

Several lines of evidence demonstrate the interplay between ECs, innate immunity, lipid metabolism and endothelial dysfunction. ECs regulate lipid metabolism through the uptake of circulating fatty acids (FAs) via receptors expressed on the EC surface [153]. Intracellularly, the FAs are esterified to triglycerides and stored in lipid droplets to be hydrolyzed as an energy source or maintain other cellular functions such as cell signalling and membrane synthesis [154]. One of the key EC receptors involved in fatty acid uptake is the CD36 receptor [155], a member of the class B scavenger receptors [155]. CD36 functions as a PRR [20]. Excess lipid droplets stimulate the NF-kB pathway, induce MCP-1 [156] and reduce eNOS expression [156].

Another common lipid metabolite that is recognized by EC surface receptors is oxLDL [157], which is generated by nicotinamide adenine dinucleotide phosphate (NADPH) oxidase [158], lipoxygenase (LOX) [159] and myeloperoxide (MPO) [157]. OxLDL binds to the scavenger receptor class B type 1 (SR-B1) [160], activin-receptor like kinase 1 (ALK1) [161], LOX-1 [21] and TLRs on ECs to induce the expression of adhesion molecules. In addition, oxLDL uncouples eNOS and attenuates NO release through activation of the RhoA/ROCK pathway [162].

Lysophospholipids, a class of proinflammatory lipid modulators derived from oxidative reactions or hydrolysis by phospholipase [163], contribute to endothelial dysfunction in a similar manner as oxLDL [164]. In addition, repeated exposure of ECs to lysophospholipids promotes the transdifferentiation of ECs with innate immune functions, leading to sustained EC activation [164].

Lipoprotein(a) [Lp(a)] is increasingly recognized as a causal risk factor for atherosclerotic cardiovascular disease [165]. Characterized by the irreversible binding of oxidized phospholipids to apo(a), which is bound to the apoB100 of LDL, Lp(a) is more atherogenic than LDL [166], potentially with greater propensity to trigger endothelial inflammation [166]. In vivo studies incubating healthy monocytes with Lp(a) and unstimulated ECs demonstrated a doubling of monocyte adhesion rate to the endothelium, accompanied by a 5-fold increase in transendothelial migration of monocytes [167]. Transcriptomics analyses of ECs exposed to Lp(a) showed upregulation of genes involved in leukocyte migration [168].

Repeated exposure to DAMPs can trigger hyper-responsive inflammatory responses. This emerging concept, known as trained immunity, describes the expansion of immunological memory beyond adaptive immunity [169]. While primarily described in innate immune cells, accumulating evidence indicates the induction of trained immunity in human ECs by DAMPs via PRRs [170]. In vitro incubation of human aortic ECs with oxLDL and subsequent re-exposure several days later increased IL-6, IL-8, MCP-1 and endothelial adhesion molecules [170]. Furthermore, there is a close association between trained immunity and metabolic changes in the ECs, in which the activation of the mammalian target of rapamycin (mTOR) pathway drives a shift towards glycolysis [170]. Furthermore, histone H3K27a and H3K4me3 activation at the ICAM-1 promoter region [170] was reported to induce trained immunity in ECs. Consequently, the administration of methyltransferase inhibitor to ECs blocked the induction of trained immunity, suggesting a possible role of epigenetic modifications [170].

## 8. Infectious Diseases and Endothelial Dysfunction

The coronavirus disease 2019 caused by the severe acute respiratory syndrome virus 2 (SARS-CoV-2) demonstrated the profound impact of infections on the endothelium [171]. Direct invasion of the SARS-CoV-2 virus into endothelial cells [172] induces massive release of proinflammatory cytokines, causes persistent endothelial injury, and increases endothelial permeability to inflammatory cells [173]. The Influenza A virus is another common respiratory virus that induces endothelial dysfunction. A recent observational study of more than 26,000 individuals with laboratory-confirmed influenza A reported a 16 times increased incidence of acute myocardial infarction within 1 year of infection among individuals without prior coronary artery disease or elevated cardiovascular risk [174]. The human immunodeficiency virus (HIV) is another independent risk factor for ASSCVD with endothelial dysfunction serving as a crucial link to atherogenesis [175]. Specific glycoproteins such as gp120 and Tat, increase EC permeability and promote EC apoptosis [176]. Another virus that preferentially invades ECs is the cytomegalovirus (CMV) [177]. CMV promotes lipid uptake in macrophages by upregulating the scavenger receptor CD36 [178], enhances the release of angiogenic factors [179], and induces EC apoptosis [180]. In terms of adaptive immunity, CMV increases CD4+ and CD8+ T cells, which could lead to plaque buildup and rupture [181]. Apart from viruses, certain bacteria species such as Chlamydia pneumoniae directly invade ECs, triggering the release of inflammatory cytokines and adhesion molecules [182]. This evidence highlights the importance of eliciting a detailed medical history, as exposure to these infectious agents could enhance cardiovascular risk.

## 9. Environmental Factors and Endothelial Function

There is growing awareness of the impact of environmental influence on endothelial function. An emerging area of research focus is the exposome, a concept that defines the total exposure to non-traditional risk factors in contemporary societies including environmental pollution and social stressors [183]. In a way, the pollutants serve as DAMPs, binding to TLRs to mediate endothelial dysfunction [184]. Long-term exposure to high levels of air pollution may destabilize atherosclerotic plaque through increasing the necrotic core [185]. Chronic exposure to social stressors is associated with endothelial dysfunction as evidenced by the elevated levels of endothelial adhesion molecules and reduced brachial artery flow-mediated dilation (FMD), a clinical marker of endothelial dysfunction [186].

We reviewed the key players in the immune–inflammatory axis by referencing prevalent metabolic disorders: hyperglycemia, obesity, hypertension, dyslipidaemia and highlighted emerging risk factors: specific infectious diseases and environmental factors (Figure 2).

As an early manifestation of cardiovascular disease, the ability to detect endothelial dysfunction in the absence of overt cardiovascular complications is crucial. In the following section, we discuss potential methods of assessing endothelial dysfunction, focusing on the biochemical and physiological markers.

Given the heterogeneous functions of the endothelial cell and various mechanisms that may lead to endothelial dysfunction, characterization of the dominant pathways involved could be useful in improving cardiovascular risk prediction beyond traditional risk factors (e.g., glucose and lipid profiles). There is heightened interest in measuring circulating biomarkers for the assessment of endothelial dysfunction and cardiovascular disease. One of the most commonly studied biomarkers is the high-sensitivity C-reactive protein (hs-CRP). In a meta-analysis comprising more than 200,000 middle-aged adults without a prior history of atherosclerotic cardiovascular disease (ASCVD), one standard deviation increase in hs-CRP increased the risk of ASCVD by 17% [187]. An analysis of 3 contemporary trials (STRENGTH, PROMINENT, and REDUCE-IT) of more than 30,000 high-risk patients on statins found that residual inflammation, determined by hs-CRP level, outweighed residual cholesterol in predicting cardiovascular events [188]. Other inflammatory markers were evaluated in large cohorts. The MESA study reported an association between IL-6, all-cause and cardiovascular mortality in individuals without baseline ASCVD [189]. This finding was corroborated by the Atherosclerosis Risk in Communities cohort, which also demonstrated an association between IL-18 with ASCVD independent of cardiovascular risk factors and hs-CRP [190]. These findings strengthen the potential role of incorporating other inflammatory markers besides hs-CRP, given the additional prognostic information that could be derived.

Other aspects of endothelial dysfunction can be assessed by profiling other endothelial markers such as the adhesion molecules, E-selectin, ICAM-1, and VCAM-1. Prospective observational studies in different cohorts in the general population demonstrated associations between circulating adhesion molecule concentrations with atherosclerotic cardiovascular disease, congestive cardiac failure and cardiovascular mortality [191,192,193,194]. Circulating markers of EG shedding, including syndecan-1, endocan and hyaluronan, are higher in individuals with cardiovascular risk factors [39] and established ASCVD [195]. EG biomarkers provide a quantitative assessment of the endothelial glycocalyx status while emerging techniques such as intravital microscopy and sidestream darkfield imaging allow direct visualization of the perfused boundary region, a surrogate marker of EG thickness [196].

The clinical assessment of endothelial function can be assessed via several methods. Coronary angiography provides direct access to the coronary endothelium, allowing vasodilatory response of coronary arteries to vasoactive agents such as acetylcholine to be measured directly [197]. However, the highly invasive nature of this procedure precludes its implementation in routine clinical settings. A non-invasive procedure, FMD, with good correlation with coronary angiography [198] in assessing endothelial function was developed in 1992. This procedure uses ultrasound to assess changes in brachial artery dilation as a surrogate marker of endothelial NO release and during reactive hyperaemia induced by sustained pressure [199].

Early detection of endothelial dysfunction provides a window of opportunity for intervention. We described blood-based and imaging markers to assess endothelial function. While these biomarkers were evaluated in limited cross-sectional studies demonstrating associations with ASCVD, several challenges limit their clinical implementation. Firstly, validating the incremental value of these endothelial-specific biomarkers is lacking in real-world populations. Large contemporary cohorts with adequate follow-up will be required to demonstrate the benefits of incorporating these biomarkers in cardiovascular risk stratification. Secondly, well-designed randomized controlled trials demonstrating the clinical utility of these biomarkers, for example, in patient selection for specific interventions, are required. Thirdly, the methodologies need to be standardized and population-specific reference values should ideally be developed [200]. For example, FMD has yet to be implemented in clinical settings because of heterogeneous measurement protocols with widely variable adherence to contemporary measurement techniques affecting the reproducibility of FMD data [197,201]. With the curation of biobanks globally, large prospective population-specific cohorts are followed up to provide opportunities to enhance our understanding of the clinical utility and the incremental value of assessing endothelial biomarkers in predicting cardiovascular events, considering interindividual differences in endothelial phenotypes. Deep profiling of endothelial biomarkers could serve as additional tools to improve risk stratification, particularly in intermediate-risk individuals whose risk may be underestimated [202]. Increased adherence to international consensus guidelines to standardize the FMD procedure would be a key step to facilitating the comparison of FMD data across centres and paving the way for clinical implementation of FMD [200].

Table 1 summarizes selected biomarkers reflecting the different aspects of endothelial function.

In the following section, we appraise recent, repurposed novel and early-phase therapeutic strategies to modulate the immune–inflammatory axis in endothelial dysfunction.

## 10. Therapeutic Strategies in Modulating the Immune–Inflammatory Axis in Endothelial Function

### 10.1. Anti-Diabetic Agents

#### 10.1.1. Sodium-Glucose Cotransporter-2 (SGLT2) Inhibitors

Although glucose lowering was the primary therapeutic goal, SGLT2 inhibitors are increasingly used to treat a wider range of diseases given their cardio-renal protective effects [205]. SGLT2 channels, located in the renal tubules, are responsible for reabsorbing approximately 90% of glucose filtered in the glomeruli [206]. The cardio-renal benefits of SGLT2 inhibitors have been extensively investigated. At the endothelial level, SGLT2 inhibition improved endothelial-dependent vasorelaxation in animal models, partly by ameliorating oxidative stress and restoring NO bioavailability [207]. In addition, SGLT2 inhibitors reduce adhesion molecule expression and leukocyte adhesion in mouse ECs [207]. The effect of SGLT2 inhibitors on suppressing NLRP3 inflammasome formation could be observed within 1 month [208]. This process is partly driven by increased ketone bodies [208,209]. The modulation of TLR4 expression by SGLT2 inhibition [210] alters the innate immune profile by enhancing the transition of M1 to M2 macrophages [211]. In terms of adaptive immunity, SGLT2 inhibitors empagliflozin enhanced the function of Treg cells while inhibiting proinflammatory Th17 cells, while canagliflozin interfered with T-cell receptor signalling and prevented T-cell mediated inflammation [212]. In humans, the DEFENCE trial was one of the first to assess the effect of SGLT2 inhibition on vascular function [213]. Daily use of dapagliflozin 5 mg over 16 weeks improved flow-mediated dilation (FMD) of the brachial artery, the gold standard clinical marker of endothelial function [213]. A meta-analysis subsequently confirmed improvement in FMD, likely a class effect of SGLT2 inhibitors [214].

#### 10.1.2. Glucagon-like Peptide-1 (GLP-1) Receptor Agonist (GLP-1 RA)

GLP-1 RAs are another class of anti-diabetic agent that has cardiorenal benefits [215]. The atheroprotective mechanisms include promoting macrophage polarization to the M2 phenotype [216,217] while suppressing M1 macrophage polarization through activating STAT 3 and inhibiting STAT 1 [217,218]. GLP-1 receptors are also expressed on human neutrophils, and their activation can significantly reduce cytokine production [219]. In terms of adaptive immunity, GLP-1 RAs inhibit the secretion of IFN-γ by natural killer (NK) cells [220] and enhance Treg function to promote IL-10 expression [221]. Additional protective effects of GLP-1 RAs include impairing monocyte adhesion to ECs through regulating the Kruppel-like factor 2 [222] and increasing eNOS activity through activating the cyclic AMP-dependent protein kinase A and PI3K-Akt pathways in ECs [223].

#### 10.1.3. Glucose-Dependent Insulinotropic Polypeptide (GIP)-GLP1 Receptor Coagonist

Tirzepatide, a novel dual GIP/GLP-1 receptor coagonist approved by the FDA in November 2023 [224], reduced HbA1c by approximately 19% [225] and body weight by 15% at the highest dose [226]. Preclinical studies indicate that tirzepatide reduced inflammatory responses, as evidenced by reductions in proinflammatory cytokines TNF-α, IL-6 and IL-1B in mice [227]. Furthermore, post hoc analyses of human clinical trial data showed a reduction in endothelial activation molecules ICAM-1 and VCAM-1 [228]. Tirzepatide also ameliorates visceral adipose tissue inflammation by inhibiting the expression of inflammation-related genes, suppressing extracellular signal-regulated kinase (ERK) signalling and reducing the infiltration of proinflammatory M1 macrophages into the adipose tissue of obese mice [229].

### 10.2. Lipid-Lowering Agents

#### 10.2.1. Statins

Statins exhibit several immune-mediated mechanisms that could mitigate the atherosclerotic process [230]. These include reducing the recruitment of monocytes into the vascular intima [230], inhibiting MHC II expression on macrophages and ECs, thereby impairing the antigen-presenting function of these cells and reducing T cell recruitment [230].

#### 10.2.2. Icosapent Ethyl

Icosapent ethyl [eicosapentaenoic acid (EPA)] is approved in international guidelines for high-risk patients with hypertriglyceridemia on statins [231]. The REDUCE-IT trial reported a 25% reduction in cardiovascular events with EPA use [232]. EPA exerts several protective mechanisms, including stabilizing cell membranes, inhibiting lipid oxidation and downregulating proinflammatory pathways such as NF-κB [233]. In addition, EPA limits the differentiation of T cells into the proinflammatory Th1 subtype [234], while promoting atheroprotective Treg cell differentiation [235]. Furthermore, EPA has been associated with reduced cytokine IL-6 [236]. EPA can also be metabolized into the lipid mediator resolvin E1, which reduces oxLDL uptake by macrophages and binds to the specific resolvin receptor ChemR23, expressed on ECs, to mediate inflammation [237].

#### 10.2.3. Lp(a) Lowering Agent

Lp(a) is a potent driver of endothelial inflammation. Targeted Lp(a) lowering therapies are being evaluated in clinical trials. These include pelacarsen (AKCEA-APO(a)-LRx), an antisense oligonucleotide, which reduced Lp(a) levels by 80% with bi-weekly administration over 6 months, compared to a 6% reduction observed with a placebo [238]. Notably, pelacarsen also reduced inflammatory gene expression and transendothelial migration of monocytes [239].

#### 10.2.4. Colchicine

Commonly used as a pharmacological treatment for gout flares, colchicine was recently repurposed for use in cardiovascular disease. In 2023, FDA approved colchicine as the first targeted anti-inflammatory agent for atherosclerotic cardiovascular disease [240]. Several clinical trials demonstrated the benefits of colchicine in secondary prevention. The colchicine cardiovascular outcome (COLCOT) and LoDoCo2 trial evaluated the addition of low-dose (0.5 mg) colchicine versus a placebo to optimize medical therapy for the secondary prevention of major cardiovascular events [241,242]. After a median follow-up of approximately two years, the colchicine group had a 23% reduction in major cardiovascular events compared to the placebo when colchicine was initiated among individuals who had suffered from an acute myocardial event within the preceding 30 days [241]. In another study, cardiovascular events were reduced by 31% among individuals with pre-existing coronary artery disease after approximately 29 months of low-dose colchicine [242].

Microtubules are integral components of the cytoskeleton, maintaining cellular architecture, regulating cellular trafficking, inflammatory mediator production and mitosis [243]. By binding to tubulin, colchicine interferes with microtubule formation and alters downstream leukocyte functions [244]. One of the main mechanisms underpinning the anti-inflammatory properties of colchicine is the inhibition of NLRP3 inflammasome formation, leading to reduced interleukin-1β and IL-18 production and limiting the transformation of macrophages into foam cells [245]. In addition, colchicine modulates the immune landscape. A study investigating the impact of colchicine on immune cellular subsets in obese individuals found that 12 weeks of colchicine exposure reduced classical monocytes, NK cells, dendritic cells, CD4+ and CD8+T cells [246]. Colchicine inhibits neutrophilic chemotaxis [247], reduces the release of activated neutrophil products such as MPO [248], blocks the formation of neutrophil extracellular traps (NETs) [249], thereby limiting the formation of atherothrombotic neutrophil–platelet aggregates [250].

### 10.3. Monoclonal Antibodies

#### 10.3.1. Canakinumab (IL-β Inhibitor)

The CANTOS trial assessed the role of monoclonal antibodies in modulating the immune system in humans against IL-1β. In this randomized controlled trial involving more than 10,000 patients with a history of acute myocardial infarction and high-sensitivity C-reactive protein (hsCRP) levels greater than 2 mg/L, canakinumab, a monoclonal antibody against IL-β, administered over a median of 3.7 years was associated with a 15% reduction in recurrent cardiovascular events compared to a placebo [251]. Inhibition of IL-1 receptors reduces the expression of endothelial adhesion molecules and blunts the proinflammatory effects induced by IL-1β [252].

#### 10.3.2. Tocilizumab (IL-6 Inhibitor)

IL-6 is a proinflammatory cytokine that enhances monocyte activity, thrombocytosis, endothelial permeability and attenuates endothelial-dependent vasodilation [253,254]. Subgroup analysis of the CANTOS trial discovered that a greater reduction in IL-6 was associated with a greater reduction in adverse cardiovascular events versus those with no change in IL-6 [255]. In the ASSAIL-MI trial, a single dose of tocilizumab administered to patients within 6 h of the onset of ST-elevation myocardial infarction (STEMI) salvaged the myocardium and ameliorated microvascular obstruction 3 to 7 days post-treatment [256]. In another randomized controlled trial, patients with non-ST elevation myocardial infarction who received tocilizumab had lower hsCRP and troponin T, suggesting a potential role of tocilizumab in attenuating the inflammatory response in acute myocardial infarction [257]. To study the expanded use of IL-6 inhibition, RESCUE, a phase II trial that recruited 264 individuals with an hsCRP of at least 2 mg/L and with moderate to severe chronic kidney disease reported that the administration of IL-6 inhibitor, ziltivekimab, over 24 weeks, was associated with nearly 88% reduction in hsCRP [258]. Ziltivekimab is currently under investigation in the ZEUS trial, which recruited more than 6000 patients with ASCVD, CKD stage 3–4 with hsCRP greater than 2 mg/dL to assess potential benefits in reducing recurrent ASCVD and the progression of renal disease [259].

#### 10.3.3. IL-2 Analog

In a preclinical phase 1b/2a trial, low-dose recombinant IL-2, aldesleukin, was administered daily for 5 days in patients with stable ischaemic heart disease and acute coronary syndrome [260]. For both groups of patients, aldesleukin increased Treg cells by 75% without significant adverse events [260]. Further analysis using single-cell sequencing found 30 cell types, including CD16- NK cells, nonclassical monocytes and CD8+ T effector memory cells, were reduced [260]. Furthermore, there was a dose-dependent effect of aldesleukin on T cell metabolism, in which a higher dose was associated with increased Treg cell glycolysis, whereas a lower dose preferentially increased oxidative phosphorylation, an effect that is postulated to suppress inflammation [260].

### 10.4. Potential Immunotherapies Targeting Other Inflammatory Mechanisms

#### 10.4.1. NLRP3 Inflammasome Inhibitor

Apart from colchicine, which inhibits NLRP3 formation non-selectively, other specific NLRP3 inflammasome inhibitors are currently in development and undergoing preclinical testing. The MCC950 is a potent NLRP3 inhibitor with preliminary evidence demonstrating efficacy in inhibiting myocardial fibrosis, limiting myocardial infarct size in animal models [261] and protecting against cardiac dysfunction [262]. As a targeted NLRP3 inhibitor, a 7-day treatment of MCC950 reduced neutrophil infiltration and IL-1β expression [260]. In angiotensin-II-induced hypertensive mice, MCC950 lowered IL-1β, improving myocardial function [263]. A novel anti-NLRP3 inflammasome antibody, InflamAb, was recently evaluated in an in vitro study involving treating macrophages with InflamAb, effectively inhibiting IL-1β release in Western diet-fed mice [264] and inhibited atherosclerotic plaque development.

#### 10.4.2. cGAS-STING Inhibitor

As an inflammatory mediator of IFN-1 signalling, STING drives systemic inflammation and is closely linked with cardiovascular diseases [107]. The STING pathway has emerged as a key therapeutic target, and several STING inhibitors are currently being tested in clinical trials. Astin C, a natural peptide, was identified as a potent inhibitor of the cGAS-STING signalling by binding to the C terminal activation pocket of STING to prevent the activation of the STING signalosome [265]. This compound can reduce IFN-1 expression in a dose-dependent manner [265]. Another mechanism of reducing STING activation is to selectively block the binding of cGAS to DNA [266]. Hydroxychloroquine is an example of a STING inhibitor that interferes with forming the cGAS-DNA complex, which is essential to activate the STING pathway [266]. Although not repurposed as a cardiovascular drug, hydroxychloroquine was shown to reduce cardiovascular risk in patients with rheumatoid arthritis and systemic lupus erythematosus [267,268]. Some of the protective mechanisms of hydroxychloroquine include improving endothelium-dependent vasodilation, reducing oxidative stress, targeting TLR signalling to modulate cytokine production and T-cell activity [269] and endothelial adhesion molecules [270]. C-176, a STING inhibitor, slowed atherogenesis in apoE null mice [118], alleviated STING-induced endothelial injury by inhibiting the IRF3/NF-κB pathway, making STING inhibitors potential therapeutic agents against vascular inflammation [107]. C-176 has also been shown to modulate the endothelial-immune axis by regulating chemotaxis, leukocyte adhesion and macrophage infiltration [271]. Another nitrofuran-derived STING inhibitor, H-151, reduced CXCL10 expression and myocardial scarring post-myocardial infarction in mice [272].

Currently, a few co-stimulatory and co-inhibitory molecules are in the early drug development phase [249]. One of the co-stimulators of T-cell activation is the CD40-CD40L complex [273]. A member of the TNF receptor family, CD40, is expressed on a broad spectrum of cell types [55]. Upon binding to the CD40 ligand, several downstream processes are activated in various immune cell subsets, promoting the differentiation of monocytes into macrophages, inducing the synthesis of proinflammatory cytokines, and activating ECs [55]. CD40L deficiency attenuates endothelial dysfunction in a mouse model exposed to continuous angiotensin-II infusion [274]. In this context, targeting the CD40-CD40L interactions may provide a novel strategy for attenuating endothelial dysfunction [275]. However, general CD40L blockade is associated with severe immunosuppression and increased thrombosis due to the disruption in the interactions between CD40L and platelets [275]. Targeted CD40-CD40L inhibitors are currently being developed. A CD40-TRAF6 inhibitor, 6877002, was administered to apoE-deficient mice for 6 weeks. The treatment reduced atherosclerotic plaque area by nearly half and reduced macrophage infiltration [276]. The overall leukocyte parameters and immune cell distribution of the treated mice were maintained, suggesting a lack of immunosuppression [276].

As described earlier, the PD-1/PDL1 axis plays an important coinhibitory role in T cell activation [58]. Checkpoint inhibition of PD-1, a therapeutic strategy employed in treating malignancies, was associated with an increased risk for atherothrombotic events [277]. Increasing PD-1 activity using PD-1 agonist in LDL-receptor deficient mice slowed atherogenesis by reducing the number of proinflammatory CD4+, CD8+ T cells and increasing the anti-inflammatory IL-10+ producing T cells [278]. These preclinical evidence supports the potential role of stimulating PD-1 as a strategy to treat atherosclerosis.

A summary of the recent, repurposed and early phase therapeutic strategies targeting the immune-inflammatory axis of ECs is provided in Table 2.

We provided an overview of the potential therapeutic agents that can be repurposed or developed to target the immune–inflammatory axis in endothelial dysfunction and atherosclerosis. In the following section, we discuss emerging areas of interest and propose how advancements in current methodologies could provide novel ways to understand individual responses to therapeutic targets of endothelial dysfunction at a personalized level.

## 11. Future Directions

In the past two decades, advancements in next-generation sequencing created unprecedented opportunities to implement precision and personalized approaches to prevent cardiovascular disease. For example, the detection of clonal haematopoiesis of indeterminate significance has been leveraged to finetune risk stratification tailor therapeutic targets. This observation was first reported in a substudy of the CANTOS trial, in which individuals carrying TET2 mutation had a 62% reduction in major cardiovascular events with canakinumab [279], suggesting the potential of implementing a precision-based approach to administer targeted and effective therapies in clinical practice.

Immune and vascular metabolism are emerging areas of interest for developing novel treatment approaches for atherosclerotic cardiovascular disease. There is a bidirectional relationship between immune function and metabolism. Alterations in the innate and adaptive immune systems contribute to the onset of metabolic disorders such as obesity and insulin resistance [280]. Similarly, changes in the metabolic pathways regulating macrophage functions can lead to detrimental downstream effects when exposed to inflammatory stimuli such as oxidized lipids and hyperglycemia [281]. OxLDL suppresses oxidative phosphorylation and shifts the intracellular metabolism toward glycolysis, a process associated with proinflammatory effects [281]. Comparable observations were noted in EC metabolism, where an increase in glycolysis promotes inflammation, abnormal angiogenesis and atherogenesis [282]. Rewiring endothelial metabolism by inhibiting glycolysis represents a potential strategy to reduce endothelial inflammation. Administration of PFK158, a glycolysis inhibitor to Lp(a) incubated ECs reduced transendothelial monocyte migration and ameliorated endothelial inflammation [167]. Similarly, Lp(a) inhibitors may potentially alter endothelial metabolism by reducing intracellular glycolysis [167]. Mechanistic studies evaluating the endothelial metabolism changes would complement the ongoing HORIZON randomized controlled trial, which assesses the reduction in major cardiovascular outcomes with Lp(a) inhibitor, pelacarsen [283].

Targeting trained immunity using nanobiotechnology platforms is a promising strategy to target the regulation of myeloid cells [284]. Nanobiologics containing the mTOR inhibitor rapamycin were tested in apoE-deficient mice and showed a significant reduction in macrophage infiltration in atherosclerotic plaques [284]. In human monocytes, beta-glucan (a DAMP) induced trained immunity in vitro and treatment with an mTOR inhibitor reduced epigenetic markers on promoters of proinflammatory cytokines and glycolysis [285].

The development of novel therapeutics targeting immune and inflammatory pathways has advanced the understanding of endothelial dysfunction and atherosclerosis. However, the fundamental principle of personalized medicine requires the delivery of the right drug to the right patient at the right time. Precision immunophenotyping will be a key area to facilitate personalized medicine. Advancements in single-cell sequencing and multi-omics technologies will continue to enhance our knowledge of the dynamic interactions between endothelial, immune, and inflammatory cell subpopulations in driving atherosclerosis [286]. Ongoing efforts to curate EC atlases from murine to human tissues provide detailed biological references to elucidate the heterogeneous functions and interactions between different endothelial cellular subsets [287]. Furthermore, there is emerging interest in characterizing endothelial cells with an organotypic perspective given the potential modulation of endothelial functions in different organ systems [288]. At the translational level, harnessing metabolomics can reveal alterations in the specific metabolic pathways, for example, glutamine metabolism, that can accelerate endothelial dysfunction and increase plaque instability [289,290]. A potential translation lies in the promising role of glutamine supplementation in mitigating atherosclerosis [291]. Another emerging area of interest is the effect of shear stress on endothelial cell function. Single cell analyses demonstrated unique transcriptomics signatures in endothelial cells exposed to disturbed shear stress [292]. Enolase 1, a key glycolytic gene, was found to be one of the top ranked mediators of endothelial phenotypic change [284], highlighting the contribution of omics research in identifying novel therapeutic targets.

Rapid immunophenotyping with microfluidic technologies could enable point-of-care testing for immune and inflammatory markers in clinical settings for real-time cardiovascular risk profiling [293]. In addition, microfluidic models of atherosclerosis-on-a-chip provide an in vivo microenvironment that reconstructs the inflammatory processes in the vessel wall, allowing real-time visualization of the effects of novel therapeutic agents on the endothelium [294]. Such microfluidic systems that mimic human vessels allow the real-time observation of dynamic structural and functional changes in endothelial cells in response to various physical stimuli [295]. Furthermore, targeted metabolomics approaches can be integrated on microvessles-on-a-chip platforms to track metabolites in response to physical stimuli. Utilizing tracer-based metabolomics, NO-specific metabolites can be tracked and quantified as surrogate markers of eNOS activity to shear stress generated by microfluidic pumps in the microvessels [296], providing the platform for comprehensive assessment of an individual’s endothelial response in physiological and pathophysiological states.

## 12. Conclusions

This review provides an overview of the interplay between immunity, inflammation, endothelial dysfunction, and atherosclerosis. Recent discoveries of novel pathways laid the groundwork for developing novel agents or repurposed treatments to modulate endothelial function and mitigate atherosclerosis progression. Precise immunophenotyping and inflammatory marker profiling can be performed almost in real-time, potentially developing personalized risk markers as point-of-care tests. Future collaborations among the scientific, industrial, and medical communities will be crucial in applying these insights and translating them into the clinical implementation of personalized risk profiling and therapies at a wider scale.

## Figures and Tables

**Figure 1 ijms-26-01708-f001:**
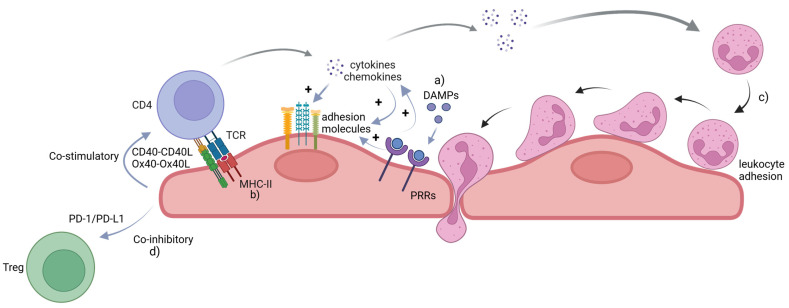
Immune–inflammatory interactions in the endothelium. The EC plays a pivotal role in modulating immunity and inflammation. With innate immune functions, ECs express PRRs that bind to DAMPs such as oxLDL, glucose, and stimulate the expression of adhesion molecules, proinflammatory cytokines, and chemokines (**a**). ECs also express MHC-II that interact with T cells to promote inflammation, a process that can be enhanced by interactions between co-stimulatory molecules on ECs and T cells, respectively (**b**). The activated endothelium attracts leukocytes including monocytes, neutrophils, and T effector cells, which undergo a series of coordinated steps to enter the subendothelial space (**c**). There are co-inhibitory interactions between ECs and T cells, which can be atheroprotective (**d**). Created in https://BioRender.com.

**Figure 2 ijms-26-01708-f002:**
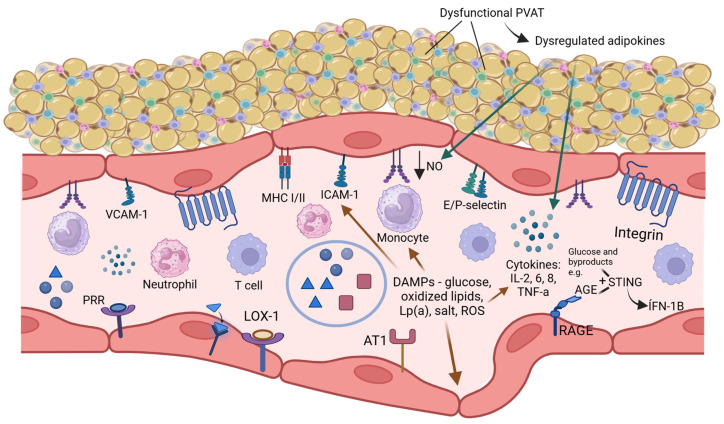
Integrated view of the immune–inflammatory axis in the endothelium. Various pathways occur in metabolically deranged states to induce inflammation in the ECs. Excess metabolites such as glucose, oxidized lipids, Lp(a), and salt can be detrimental by triggering the innate immune functions of ECs, enhancing the expression of adhesion molecules, causing the secretion of proinflammatory cytokines, promoting leukocyte migration, and damaging the endothelial barrier. In hyperglycemia, a few mechanisms including AGE-RAGE interactions, alteration of immune landscapes, activation of STING via glucose and glucose byproducts such as AGEs, as well as endothelial insulin resistance contribute to endothelial dysfunction. In obesity, excess perivascular adipose tissues become infiltrated with immune cells and become dysfunctional. Dysregulated secretion of adipokines such as leptin and visfatin on the endothelium promote inflammation further. In hypertension, innate and adaptive immune mechanisms contribute to its pathogenesis while the interactions between oxidized lipids, lysophospholipids, Lp(a) with ECs activate the innate immune responses and promote transendothelial leukocyte migration. Created in https://BioRender.com.

**Table 1 ijms-26-01708-t001:** Endothelial function biomarkers.

Biomarker	Measurement Method
Inflammation	
hs-CRP	Immunoassay [200], ELISA [203]
IL-6/IL-18	ELISA [190]
Adhesion molecules	
E-selectin	ELISA [191]
VCAM-1/ICAM-1	ELISA [191,194]
Endothelial glycocalyx	
Syndecan	ELISA [49]
Endocan	ELISA or immunoassay [204]
Hyaluronan	ELISA [43]
Non-invasive clinical measurement	
Endothelial NO availability	Flow mediated dilation [200]

**Table 2 ijms-26-01708-t002:** A summary of the recent, repurposed, and early phase therapeutic strategies targeting the immune–inflammatory axis of ECs.

Therapeutic Agent	Protective Mechanisms
SGLT2-inhibitors	Reduces the expression of adhesion molecules and leukocyte adhesion on ECs Suppresses NLRP3 inflammasome formation Modulates TLR4 Decreases the release of proinflammatory cytokines Promotes the polarization of M1 to M2 macrophages Enhances Treg cells and interfere with TCR signalling Triggers immunometabolic changes in macrophages
GLP-1 RAs	Impairs monocyte adhesion to ECs Promotes polarization of M1 to M2 macrophages Reduces cytokines release from neutrophils Inhibits secretion of IFN-y by NK T cells Enhances Treg cells to produce IL-10 Increases eNOS activity through Sirtuin 6 expression
Tirzepatide	Reduces DAMP-induced inflammation Reduces endothelial adhesion molecules
Statins	Reduces monocyte recruitment to endothelium Inhibits MHC-II expression on macrophages and ECs
Icosapent ethyl	Limits differentiation of T cells into proinflammatory CD4+ Th1 subtype Enhances differentiation into Treg cells Reduces proinflammatory pathways, e.g., NF-kB, cytokines IL-6 Reduces oxLDL uptake
Lp(a) inhibitor	Reduces inflammatory gene expression Reduces transendothelial migration of monocytes
Colchicine	Interferes with microtubule formation and leukocyte function Inhibits NLRP3 inflammasome Alters innate and adaptive immune cell subsets Reduces neutrophilic chemotaxis Reduces release of NETs
Canakinumab (IL-1β inhibitor)	Reduces adhesion molecules Attenuates proinflammatory and pro-oxidant signalling by IL-1 beta
Tocilizumab (IL-6 inhibitor)	Studies demonstrated potential benefit in acute coronary syndrome Reduces monocyte activity, leukocyte migration and thrombocytosis
Aldesleukin (IL-2 inhibitor)	Increases Treg cells Reduces CD8+ cytotoxic T cells Alters metabolism of T cells
NLRP3 inflammasome inhibitor: MCC950, InflamAb	Reduces IL-1β expression, adhesion molecule expression, neutrophil and macrophage infiltration
cGAS-STING pathway (astin C, hydroxychloroquine, nitrofuran derivatives, C-176, H-151)	Reduces IFN and TLR signalling Reduces immune chemotaxis and leukocyte infiltration
T cell checkpoint inhibition, e.g., CD40-CD40L blockade, CD40-TRAF6 inhibitor	Reduces leukocyte migration and infiltration

## Data Availability

No new data were generated.

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
