# Peer review of "The Interplay Between Immunity, Inflammation and Endothelial Dysfunction"

_ijms, 2025, doi:10.3390/ijms26041708_

Round 1
Reviewer 1 Report
Comments and Suggestions for Authors
The review article provides a comprehensive overview of the complex interactions between endothelial dysfunction, immunity, and inflammation, particularly in the context of various metabolic disorders. The authors have effectively synthesized a vast amount of literature, presenting a well-structured and informative narrative that highlights the role of endothelial cells (ECs) as mediators of both innate and adaptive immunity.
· The introduction sets a strong foundation; however, Some sections could benefit from clearer transitions. For example, the shift from discussing endothelial dysfunction to specific disorders like diabetes and obesity feels abrupt. Adding transitional sentences may enhance readability.
· The role of specific immune cell types in the context of endothelial dysfunction could be expanded. For instance, a more detailed exploration of Tregs and their protective mechanisms would enrich the discussion.
· Proofread typographical errors and ensure consistency in terminology (e.g., "endothelial cells" vs. "ECs").
Here are some grammatical mistakes and suggestions for improvement found in the review article:
· "The endothelium is pivotal in multiple physiological processes, such as maintaining vas- cular homeostasis, metabolism, platelet function and oxidative stress." Could be "The endothelium is pivotal in multiple physiological processes, such as maintaining vascular homeostasis, metabolism, platelet function, and oxidative stress."
· "immune-inflammation axis" to "immune-inflammatory axis"
· "The progression of atherosclerosis occurs over a relatively long asymptomatic interval and may begin as early as childhood." "The progression of atherosclerosis occurs over a relatively long, asymptomatic interval and may begin as early as childhood.
· The transition between sections could be smoother. For instance, the shift from discussing NLRP3 inflammasome inhibitors to the future directions in cardiovascular disease could benefit from a connecting sentence that relates the two topics.
· Consider breaking down lengthy paragraphs into shorter ones to enhance readability. Each paragraph should ideally focus on a single idea or concept.
· In the sentence "Advancements in next-generation sequencing have created unprecedented opportunities to implement precision and personalized approaches," consider revising "have created" to "have created" for parallelism with "advancements."
· The phrase "suggesting the potential of implementing a precision-based approach to administer targeted and effective therapies in clinical practice" could be clearer with a comma before "suggesting."
· In the sentence "There is a bidirectional relationship between immune function and metabolism," consider adding a comma after "function" for clarity.
· Replace "novel approaches to treat atherosclerotic cardiovascular disease" with "novel treatment approaches for atherosclerotic cardiovascular disease" for improved clarity.
· The phrase "novel therapeutic agents or repurposing existing treatments" could be simplified to "novel agents or repurposed treatments" to avoid redundancy.
· Ensure that the tense remains consistent throughout the review. For example, if discussing past research, use the past tense consistently.
Author Response
Thank you very much for taking the time to review this manuscript. Please find the detailed responses below and the corresponding revisions in track changes in the re-submitted files.
The review article provides a comprehensive overview of the complex interactions between endothelial dysfunction, immunity, and inflammation, particularly in the context of various metabolic disorders. The authors have effectively synthesized a vast amount of literature, presenting a well-structured and informative narrative that highlights the role of endothelial cells (ECs) as mediators of both innate and adaptive immunity.
Comment 1:
The introduction sets a strong foundation; however, Some sections could benefit from clearer transitions. For example, the shift from discussing endothelial dysfunction to specific disorders like diabetes and obesity feels abrupt. Adding transitional sentences may enhance readability.
Response 1: We edited the manuscript to enhance the transitions between the subtopics. For example, we expanded the paragraph on page 5, lines 229 to 238 to improve the transition from discussing endothelial dysfunction to specific disorders.
Comment 2:
The role of specific immune cell types in the context of endothelial dysfunction could be expanded. For instance, a more detailed exploration of Tregs and their protective mechanisms would enrich the discussion.
Response 2: Thank you for the comment. We expanded the discussion on Tregs (page 4, lines 181 - 195). We included a new paragraph on humoral immunity (pages 4-5, lines 197 - 216) to discuss the significance of anti-endothelial cell antibodies.
Comment 3:
- Proofread typographical errors and ensure consistency in terminology (e.g., "endothelial cells" vs. "ECs").
Here are some grammatical mistakes and suggestions for improvement found in the review article:
- "The endothelium is pivotal in multiple physiological processes, such as maintaining vas- cular homeostasis, metabolism, platelet function and oxidative stress." Could be "The endothelium is pivotal in multiple physiological processes, such as maintaining vascular homeostasis, metabolism, platelet function, and oxidative stress."
- "immune-inflammation axis" to "immune-inflammatory axis"
- "The progression of atherosclerosis occurs over a relatively long asymptomatic interval and may begin as early as childhood." "The progression of atherosclerosis occurs over a relatively long, asymptomatic interval and may begin as early as childhood.
Response 3: We have amended the grammatical mistakes.
- The transition between sections could be smoother. For instance, the shift from discussing NLRP3 inflammasome inhibitors to the future directions in cardiovascular disease could benefit from a connecting sentence that relates the two topics.
Response: We added a connecting paragraph on page 16, lines 702 - 707 to improve the transition between sections.
- Consider breaking down lengthy paragraphs into shorter ones to enhance readability. Each paragraph should ideally focus on a single idea or concept.
Response: We focused, as much as possible, on a single concept in each paragraph. The main concept that will be discussed in the paragraph is introduced in the subheadings.
- In the sentence "Advancements in next-generation sequencing have created unprecedented opportunities to implement precision and personalized approaches," consider revising "have created" to "have created" for parallelism with "advancements."
Response: We amended the above sentence (page 16, lines 709 – 711) to improve the clarity.
- The phrase "suggesting the potential of implementing a precision-based approach to administer targeted and effective therapies in clinical practice" could be clearer with a comma before "suggesting."
Response: A comma has been added.
- In the sentence "There is a bidirectional relationship between immune function and metabolism," consider adding a comma after "function" for clarity.
Response: A comma has been added.
- Replace "novel approaches to treat atherosclerotic cardiovascular disease" with "novel treatment approaches for atherosclerotic cardiovascular disease" for improved clarity.
Response: The sentence has been amended to improve clarity.
- The phrase "novel therapeutic agents or repurposing existing treatments" could be simplified to "novel agents or repurposed treatments" to avoid redundancy.
Response: The phrase has been amended to avoid redundancy.
- Ensure that the tense remains consistent throughout the review. For example, if discussing past research, use the past tense consistently.
Response: The manuscript has been proofread to ensure consistency.
Reviewer 2 Report
Comments and Suggestions for Authors
Dear Authors,
the manuscript provides a detailed exploration of endothelial dysfunction and its interplay with immunity and inflammation. It connects fundamental science with clinical implications, which is valuable for advancing cardiovascular research.
The manuscript integrates novel insights, such as trained immunity, metabolic changes, and advanced endothelial biomarkers. These additions provide depth and align with current scientific trends and by discussing biomarkers and therapeutic interventions, the work bridges basic research and clinical applications.
While thorough, the manuscript occasionally delves into excessive detail, which might overwhelm non-specialist readers. A more concise presentation of some sections could enhance accessibility. In addition, the manuscript focuses primarily on metabolic disorders, therefore, expanding on other triggers of endothelial dysfunction, such as infections or environmental factors, could provide a broader perspective.
Some sections overlap, such as the repeated discussions on adhesion molecules and inflammatory pathways. Streamlining these areas would improve clarity.
While the manuscript discusses biomarkers and techniques, it could benefit from more emphasis on clinical validation or challenges in implementing these methods in practice. Simialrly, the section on future directions hints at promising areas like multi-omics and microfluidic technologies but lacks detailed elaboration or examples of potential studies.
Last, but not least, the reference list is quite wide however looking not exhaustive, since, in reviewer, opinion there are some unlisted publications dealing with endothelial dysfunction, vascular reactivity and hyperglycaemia.
Author Response
Thank you very much for taking the time to review this manuscript. Please find the detailed responses below and the corresponding revisions in track changes in the re-submitted files.
Dear Authors,
the manuscript provides a detailed exploration of endothelial dysfunction and its interplay with immunity and inflammation. It connects fundamental science with clinical implications, which is valuable for advancing cardiovascular research.
The manuscript integrates novel insights, such as trained immunity, metabolic changes, and advanced endothelial biomarkers. These additions provide depth and align with current scientific trends and by discussing biomarkers and therapeutic interventions, the work bridges basic research and clinical applications.
Comment 1:
While thorough, the manuscript occasionally delves into excessive detail, which might overwhelm non-specialist readers. A more concise presentation of some sections could enhance accessibility. In addition, the manuscript focuses primarily on metabolic disorders, therefore, expanding on other triggers of endothelial dysfunction, such as infections or environmental factors, could provide a broader perspective.
Some sections overlap, such as the repeated discussions on adhesion molecules and inflammatory pathways. Streamlining these areas would improve clarity.
Response 1: Thank you for the comment. We streamlined some sections (e.g. section 4: Diabetes and endothelial dysfunction) by removing duplicate concepts and terms. We added 2 paragraphs on infections (pages 8 - 9, lines 382 – 403) and environmental factors (page 9, lines 405 - 414) as emerging factors of endothelial dysfunction.
Comment 2:
While the manuscript discusses biomarkers and techniques, it could benefit from more emphasis on clinical validation or challenges in implementing these methods in practice. Simialrly, the section on future directions hints at promising areas like multi-omics and microfluidic technologies but lacks detailed elaboration or examples of potential studies.
Response 2: We elaborated the challenges in implementing biomarkers and the FMD technique (pages 10-11, lines 478 – 499). For the section on future directions, we expanded the discussion on multi-omics and microfluidic technologies and provided examples of studies in these areas (pages 17-18, lines 742 – 776).
Comment 3:
Last, but not least, the reference list is quite wide however looking not exhaustive, since, in reviewer, opinion there are some unlisted publications dealing with endothelial dysfunction, vascular reactivity and hyperglycaemia.
Response 3: We expanded the references (ref: 83 to 86) to include publications on hyperglycemia, inflammation and endothelial dysfunction.
Reviewer 3 Report
Comments and Suggestions for Authors
The manuscript “The interplay between immunity, inflammation and endothelial dysfunction” by Chee et al is an interesting compilation of information on activating mechanisms of inflammation, although the relationship that endothelial cells specifically have with the damaging effect caused by the agents is not clear. various inflammatory factors in endothelial cells but with the various cellular components of the innate immune response. For example, it is monocytes that contribute various factors that cause endothelial dysfunction. There are some other factors that are not considered, such as the deposits of immune complexes from autoimmune diseases and the participation of the complement system, as well as deficiencies in regulatory factors of complement function. However, the work is interesting and will have the attention of specialists in the area.
An interesting topic raised in this work is pharmacological alternatives, which also outlines the current therapeutic alternatives to regulate the harmful effects of inflammation, which, as can be identified in the chapter, are not therapies specifically directed at endothelial damage, however, in a systemic way. They provide alternatives to avoid endothelial damage.
There are some minor aspects that the authors of this work suggest review that could cause misunderstandings in the readers of this work, e.g. L59 “The innate immune system generates immediate, non-specific immune responses” what do the authors refer to as non-specific? There is much evidence that indicates that the innate immune response, unlike the adaptive one, does not generate immunological memory, but all PRRs have specificity towards their PAMP, DAMP or LAMP type ligands. Even γδ T cells possess specificity features in the recognition of molecules presented in the context of CD-1. This idea of ​​non-specificity is decadent.
Figure 2 is very interesting, however it does not show the direct effect of the atheromatous plaque (or perivascular lipids) on the endothelium and simply summarizes a group of molecules in a circle related to DAMPS such as glucose, and the question in this topic It is whether glucose per se is a DAMP or are all the oxidative processes related to trying to regulate its concentration (such as glycated hemoglobin, etc.). Likewise, the text mentions the STING-AGE-RAGE relationship, but in the diagram they appear to be independent aspects, where the main effect of STING is to favor the production of IFN-1 in the endothelium? Or IFN-type 1 (IFNA-alpha and beta?). Perivascular adipose tissue could contribute important factors such as leptins that would have a vascular inflammatory effect.
L 122 The idea discussed in paragraph 122-123 « consequently, ECs are referred to as “semi-professional APCs” that can activate T effector memory cells with prior exposure to the presented antigen”, should be reconsidered, at least in term In general, it is known that the absence of CD80/86 or CD40/CD40L markers favors a state of cellular anergy, the authors' interest in subclassifying cells into professional and non-professional presenters. and “semi-professional” does not have an important contribution, on the contrary, trying to identify the role of the endothelium in the would not have an effect on the generation of perivascular lipid deposits?
Author Response
Thank you very much for taking the time to review this manuscript. Please find the detailed responses below and the corresponding revisions in track changes in the re-submitted files.
Comment 1:
The manuscript “The interplay between immunity, inflammation and endothelial dysfunction” by Chee et al is an interesting compilation of information on activating mechanisms of inflammation, although the relationship that endothelial cells specifically have with the damaging effect caused by the agents is not clear. various inflammatory factors in endothelial cells but with the various cellular components of the innate immune response. For example, it is monocytes that contribute various factors that cause endothelial dysfunction. There are some other factors that are not considered, such as the deposits of immune complexes from autoimmune diseases and the participation of the complement system, as well as deficiencies in regulatory factors of complement function. However, the work is interesting and will have the attention of specialists in the area.
Response 1: Thank you for the comment. We added mechanisms, including the deposition of immune complexes, contribution of the complement system and its regulatory factors on page 3, lines 99 to 119. We added a paragraph on antibodies and endothelial dysfunction on pages 4 to 5, lines 197 to 216.
An interesting topic raised in this work is pharmacological alternatives, which also outlines the current therapeutic alternatives to regulate the harmful effects of inflammation, which, as can be identified in the chapter, are not therapies specifically directed at endothelial damage, however, in a systemic way. They provide alternatives to avoid endothelial damage.
Comment 2:
There are some minor aspects that the authors of this work suggest review that could cause misunderstandings in the readers of this work, e.g. L59 “The innate immune system generates immediate, non-specific immune responses” what do the authors refer to as non-specific? There is much evidence that indicates that the innate immune response, unlike the adaptive one, does not generate immunological memory, but all PRRs have specificity towards their PAMP, DAMP or LAMP type ligands. Even γδ T cells possess specificity features in the recognition of molecules presented in the context of CD-1. This idea of ​​non-specificity is decadent.
Response 2: Thank you for the comment. We have removed the term “non-specific” (line 59).
Comment 3:
Figure 2 is very interesting, however it does not show the direct effect of the atheromatous plaque (or perivascular lipids) on the endothelium and simply summarizes a group of molecules in a circle related to DAMPS such as glucose, and the question in this topic It is whether glucose per se is a DAMP or are all the oxidative processes related to trying to regulate its concentration (such as glycated hemoglobin, etc.). Likewise, the text mentions the STING-AGE-RAGE relationship, but in the diagram they appear to be independent aspects, where the main effect of STING is to favor the production of IFN-1 in the endothelium? Or IFN-type 1 (IFNA-alpha and beta?). Perivascular adipose tissue could contribute important factors such as leptins that would have a vascular inflammatory effect.
Response 3: Thank you for the comment. We agree that it is not just glucose per se but also the other glucose metabolites such as AGEs that trigger immune-inflammatory responses in the endothelium. This change is reflected on page 6, lines 261 to 274.
We would like to clarify that STING-AGE-RAGE stimulates IFN-type 1β synthesis. We amended figure 2 to show the effect of glucose and its byproducts e.g. AGE on the STING pathway.
We expanded section 5 on obesity and endothelial dysfunction (page 7, lines 305 to 311) to include a discussion on adipokines such as leptins.
Comment 4:
L 122 The idea discussed in paragraph 122-123 « consequently, ECs are referred to as “semi-professional APCs” that can activate T effector memory cells with prior exposure to the presented antigen”, should be reconsidered, at least in term In general, it is known that the absence of CD80/86 or CD40/CD40L markers favors a state of cellular anergy, the authors' interest in subclassifying cells into professional and non-professional presenters. and “semi-professional” does not have an important contribution, on the contrary, trying to identify the role of the endothelium in the would not have an effect on the generation of perivascular lipid deposits?
Response 4: Thank you for the comment. We have removed the term “semi-professional”. (lines 144-145). We added a paragraph to discuss the interplay between ECs, innate immunity, lipid metabolism and endothelial dysfunction (page 7 – 8, lines 338 to 346).
Round 2
Reviewer 2 Report
Comments and Suggestions for Authors
Dear Authors,
I appreciated the effort to improve the manuscript, however, with respect to my comment n.3, the references added are a bit outdated and only describing cytokines release rather than discuss a wider concept as the endothelial impairment associated to vascular dysfunction.
This aspect requires a thorough attention by the authors.
Kind regards
Author Response
Thank you very much for taking the time to review this revised manuscript. Please find our responses below and the corresponding revisions highlighted in the resubmitted manuscript.
Reviewer's comment:
I appreciated the effort to improve the manuscript, however, with respect to my comment n.3, the references added are a bit outdated and only describing cytokines release rather than discuss a wider concept as the endothelial impairment associated to vascular dysfunction.
This aspect requires a thorough attention by the authors.
Authors' comment:
Thank you for the comment. We revised section 4: Diabetes and endothelial dysfunction (page 6, lines 245 to 288). We elaborated further on the mechanisms of oxidative stress (including a discussion on mitochondrial ROS), endothelial insulin resistance, and discussed the role of neutrophil extracellular traps and Takeda G Protein-Coupled Receptor 5 in endothelial dysfunction.